# Experimental Study of Three-Dimensional Time Domain of Water Evaporation from Goaf

Chuanbo Cui [1,2,*], Zhipeng Jiao [1], Yuying Zhou [1], Shuguang Jiang [2], Yanwei Yuan [1], Jiangjiang Li [1] and Zhiqiang Song [1]

[1] School of Safety and Emergency Management Engineering, Taiyuan University of Technology, Taiyuan 030024, China; jiaozpeng@163.com (Z.J.); zhouyuying0786@163.com (Y.Z.); ywy1602795332@163.com (Y.Y.); 18406588504@163.com (J.L.); szq199802@163.com (Z.S.)

[2] School of Safety Engineering, China University of Mining and Technology, Xuzhou 221116, China; jsguang@cumt.edu.cn

\* Correspondence: cuichuanbo@tyut.edu.cn; Tel.: +86-15-034123543

**Abstract:** Previous theories on coal spontaneous combustion (CSC) prevention and control mostly focused on how to improve the CSC-inhibiting effect instead of the failure of the inhibitors. Due to the influence of air leakage in the goaf, the inhibitors and the water in the coal evaporate continuously, thus making the effect of the inhibitors to inhibit coal spontaneous combustion weakened or even null. Due to the complex environment inside the goaf, it is difficult for personnel to enter and collect data such as coal moisture. Therefore, a three-dimensional model of the mining area was built to study the changes in coal moisture inside the goaf under different air leakage conditions. The analysis of the experimental results shows that the larger the air leakage volume is, the faster the moisture evaporates inside the goaf. The moisture of the coal body was reduced from 30% to about 5% in 15, 11 and 7 days when the air leakage was 0.02, 0.03 and 0.04 $m^3/s$, respectively. The law of moisture evaporation at different times and locations in the mining area was also studied and it was found that the evaporation of moisture from the coal body in the mining area was location- and time-dependent.

**Keywords:** goaf; air leakage; water evaporation; inhibitor failure; three-dimensional model

## 1. Introduction

With the social and economic development in China and the improvement in people's living standards, shallow coal seams in the central–eastern region are consumed in large quantities. As the depth of mining continues to deepen, the pressure on coal mining companies to produce safely is gradually increasing. Greater human, material and financial resources are required to prevent accidents from occurring. Among the various accidents, mine fire accidents caused by spontaneous coal combustion are more serious [1–3]. Fifty-six percent of China's coal mines have had spontaneous coal combustion fires, resulting in serious economic losses and widespread environmental pollution [4,5]. There have been various causes of mine fires, including spontaneous coal fires, electrical accidents, etc. Spontaneous coal combustion causes over 90% of all mine fires [6–8]. Fires caused by spontaneous combustion of coal can release large amounts of toxic and harmful gases ($CO$, $CO_2$, $SO_2$, etc.). Toxic and harmful gases threaten people's lives and health, pollute the environment and can trigger gas and coal dust explosions [9–13]. So far, the coal resources that cannot be used due to the spontaneous combustion of coal amount to 200 million tons [14,15]. Every year, 10~13.6 million tons of standard coal is wasted due to coal spontaneous combustion, and the estimated economic loss is CNY 20 billion [16,17].

Based on the above analysis, China is facing a very serious mine fire situation and needs to pay great attention to the prevention and management of coal spontaneous combustion disasters. However, in the past, a significant number of studies on coal spontaneous combustion prevention and control have focused on how to improve the effectiveness of the inhibitor, with little research on the failure of the inhibitor. Additionally,

almost no scholars have studied the time-domain failure of inhibitors in the goaf. In general, the main reason for the failure of the blocking agent is divided into two aspects: one is the physical role of the blocking agent which exists in the evaporation of water; inert gas easily flows and escapes, making the coal and oxygen contact area increased and accelerating oxidation; the second is that the chemical role of the blocking agent is not stable, as an anti-oxidation-type blocking agent exposed to air for a long time and resulting in failure. It is well known that the oxidation process of coal continues even at room temperature. When the heat produced by the oxidation of the coal body is greater than the heat dissipated, the coal temperature begins to rise [18,19]. When the temperature rises to the critical temperature of coal spontaneous combustion, the coal oxidation process will be accelerated [6,20,21]. However, it is a very long process for the temperature of the relic coal in the goaf to reach the critical temperature from room temperature [22]. In this process, it is generally believed that the air leakage from the goaf and the heat released from the coal oxidation will cause the water in the coal to evaporate. Therefore, the coal body sprayed with the resist solution will reduce its own moisture due to water evaporation. Additionally, a significant number of scholarly studies have shown that the evaporation of water from the coal body accelerates the process of coal oxidation [23–27]. A conventional inhibitor includes a physical inhibitor and a chemical inhibitor, such as salt aqueous solution, antioxidant class, etc. Among them, the physical inhibitor, mainly through its own highly absorbent qualities in the coal body surface, forms a liquid film to block the contact between the coal and oxygen [28]. However, due to the evaporation of water, the physical inhibitor inhibitory effect will weaken over time or even fail, reflecting the shortcomings of the life cycle [29]. Chemical inhibitors are mainly antioxidant-based inhibitors, such as anthocyanins, vitamin C and catechins. A small amount can be added to have a very good inhibitory effect. However, antioxidant inhibitors are easily oxidized in the air, so it is difficult to maintain a good inhibitory effect in the complex environment of the well for a long time. Shao Hao et al. [30] studied the effect of inert gas on coal spontaneous combustion and found that inert gas can inhibit coal spontaneous combustion by increasing the activation energy of the coal body. However, the easy-flowing and escaping characteristics of inert gas make it impossible to be preserved for a long time. With the slurry injection technology, the slurry can easily flow to the low ground level, which is difficult to cover the coal body completely. Moreover, when the water in the slurry evaporates, the fly ash and yellow mud covering the surface of the coal body will crack, causing oxygen to enter the inside of the coal body along the cracks. The foam technology can cover the high coal body, but after the water evaporates for a long time, the foam will break and weaken its inhibitory effect. Gels and colloids have the disadvantage of high production cost and small diffusion range. In this case, the problem of weakening or even failure of the inhibitor effect may lead to the spontaneous combustion of the coal body in the goaf, which in turn may cause casualties and property damage. At present, most coal mines still use a high-water-cut physical inhibitor to inhibit the spontaneous combustion of coal in the goaf. However, air leakage in the goaf can lead to water evaporation, which weakens the inhibitory effect of the inhibitor. Therefore, constructing a three-dimensional model of water evaporation in the goaf may provide effective guidance information for the use of inhibitors.

## 2. Experimental Materials and System

### 2.1. Experimental Materials

The fresh coal samples used in this experiment were obtained from the Longdong coal mine of Xuzhou Datun Coal and Power Company, China. The fresh coal samples were stored in sealed bags, taken back to the laboratory and crushed by a coal crusher to screen out the coal samples with particle sizes of 0.18~1.18 mm. Some coal samples with particle sizes of 0.18~0.38 mm were taken and then dried in a vacuum drying oven at 40 °C for 24 h. The dried coal samples were subjected to industrial analysis and elemental analysis, and the obtained data are shown in Table 1.

**Table 1.** Technical parameters of Longdong coal sample.

| Industrial Analysis (%) | | | | Title 3 | | | | |
|---|---|---|---|---|---|---|---|---|
| Mad | Ad | Vdaf | FC.ad | St.d | Odaf | Cdaf | Hdaf | Ndaf |
| 2.38 | 25.32 | 40.96 | 43.04 | 1.15 | 13.42 | 78.49 | 5.10 | 1.45 |

### *2.2. Experimental System*
### 2.2.1. Project Overview

The site is located at the 3304 general mining face of the Zoucheng City Nantun Mine in Shandong Province, China. The face is generally a monoclinic structure, with a high west–east and low south–north direction; the coal seam has a near-southeast orientation and a near-north–east tendency, with an inclination from 2° to 6°, averaging 3°. The face length of the working face is 300 m (vertical flat spacing between the two chutes), and the length of the recovery advance is 653.4 m, with a daily advance of 3.1 m. The plan diagram is shown in Figure 1. The direction of wind flow at the comprehensive mining face is 3304 Transportation channel in coal mines—3304 Working Face—3304 Return air channel in the coal mine. The coal body of this working face is prone to spontaneous combustion. The measure to inhibit the spontaneous combustion of coal is to inject slurry into the goaf.

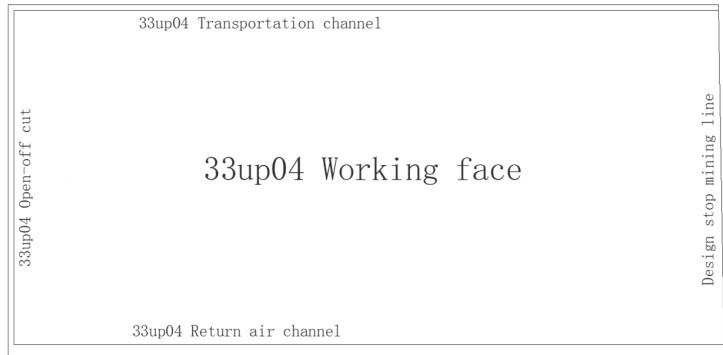

**Figure 1.** Plan view of the 3304 integrated mining face.

### 2.2.2. Three-Dimensional Models

A three-dimensional model of an experimental system for water evaporation from the inhibitor in the goaf is proposed. The system is scaled equivalently to the $33_{up}04$ integrated mining face.

The three-dimensional model experimental system of water evaporation of resistants in goaf mainly includes the water evaporation system of the inhibitors in goaf and the data acquisition system. As shown in Figures 2 and 3, the inhibitor in the goaf moisture evaporation system mainly consists of coal, experimental box, PVC pipe (20), fan (21) and fan speed regulator. The experimental acquisition system is mainly composed of the coal body moisture sensor (1~15), air temperature and humidity sensor (16, 18), Pitot tube (17, 19), handheld wind pressure and wind speed measuring instrument, monitoring substation and computer. The PVC pipe size specification is 1000 (mm) × 110 (mm).

(1) Experiment box

The internal container size specification of the experimental chamber is 1200 (mm) × 600 (mm) × 500 (mm). The box is surrounded by transparent PMMA (polymethyl methacrylate) plexiglass with high strength. The external frame of the experimental chamber is stainless steel with good stability and high strength, and the height is 0.5 m. As shown in Figure 2, the experimental chamber and lid will be sealed with glass glue to prevent air leakage before conducting the experiment.

(2) Monitoring and data acquisition system

The monitoring substation is the core of the whole monitoring and data acquisition system. The monitoring substation is divided into the following five parts according to the

function: Siemens CPU224CN, EM231 module, CP-243 network module, intrinsically safe power supply, switch, etc. The physical diagram of the site is shown in Figure 4.

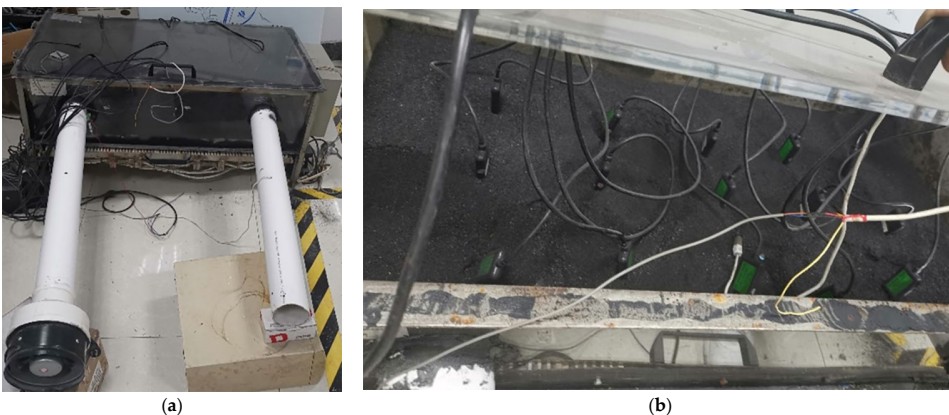

(**a**)    (**b**)

**Figure 2.** Physical diagram of the experimental system for the three-dimensional model for water vaporization of inhibitor in goaf: (**a**) Front view of the water evaporation 3D model; (**b**) Internal view of the water evaporation 3D model.

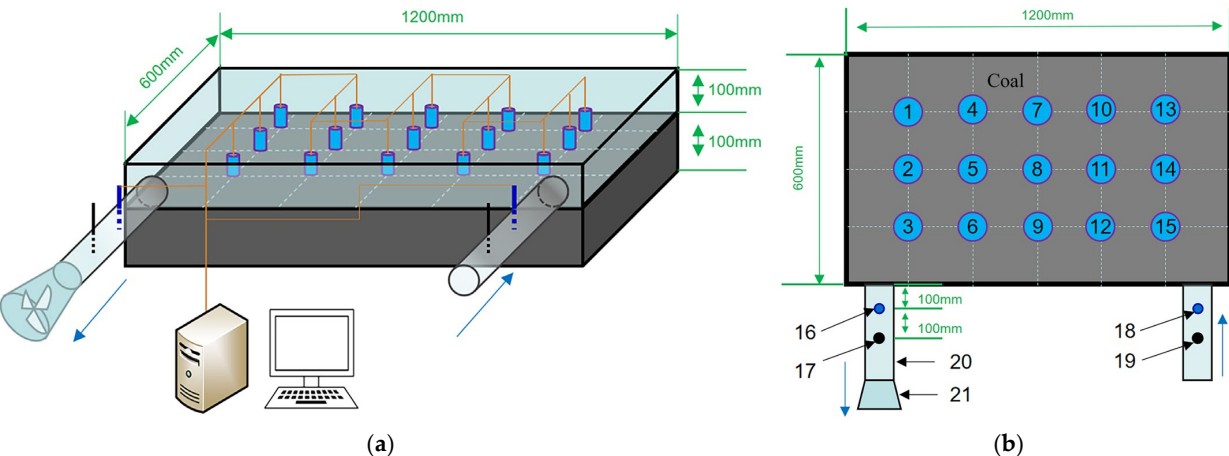

(**a**)    (**b**)

**Figure 3.** The experimental system for the three-dimensional model for water vaporization of inhibitor in goaf: (**a**) Main view of the experimental system for the three-dimensional model for water vaporization of inhibitor in goaf; (**b**) Top view of the experimental system for the three-dimensional model for water vaporization of inhibitor in goaf.

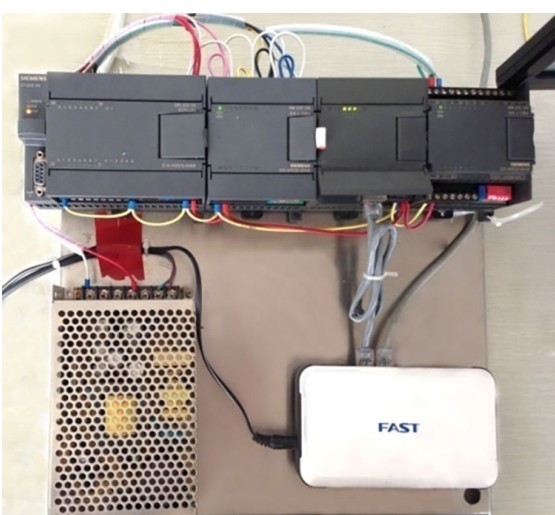

**Figure 4.** The physical figure of IP sensor substation.

### 3. Experimental Protocol

(1)    Moisture Sensor Calibration Experiment

The moisture sensor used in this paper obtains the moisture in the medium by the electrical conductivity of the medium to be measured. It is mainly used to measure the moisture in vegetable greenhouses and fruit trees. Since both soil and coal dust are in granular form, moisture is present within and between the particles. Therefore, soil moisture sensors (referred to as moisture sensors) can be used to measure moisture in granular coal bodies. However, the moisture sensor needs to be calibrated before it can be used to measure the moisture in the coal. The specific calibration steps are as follows:

- Step 1: Longdong coal samples with the particle size of 0.18~1 mm were taken and dried under a vacuum environment for 24 h.
- Step 2: Take 4 clean water buckets and pour 5 kg of dried coal samples into each of them.
- Step 3: Take three of the buckets containing coal samples and add 0.714, 1.667 and 3 kg of 20% $CaCl_2$ solution to configure the coal samples with 10, 20 and 30% moisture, respectively.
- Step 4: Insert the moisture sensor to be calibrated into the above four coal samples in turn and obtain their readings.
- Step 5: Find the relationship between the reading value and the theoretical value and perform calibration.

(2)    Water Evaporation Experiment in Goaf

Experiment steps on the effect of air leakage on the evaporation of water from the goaf:

- Step 1: Take 60 kg of dried Longdong coal samples with particle size in the range of 0.18~1 mm and lay the coal samples flat into the experimental chamber, as shown in Figure 2.
- Step 2: Insert the 15 moisture sensors into the 15 measurement point locations (M1~M15) as shown in the figure and place the two air temperature and humidity sensors at measurement points 16 and 17.
- Step 3: Connect the moisture sensor and air temperature and humidity sensor to the data acquisition device and turn on the computer for data acquisition. Turn on the stepless variable speed extension.
- Step 4: A solution of $CaCl_2$ with a concentration of 20% was evenly sprinkled into the experimental chamber to bring the moisture from M1 to M15 to 30%.
- Step 5: Use the wind pressure and air volume meter to measure the wind pressure and air velocity at the location of the 17 measuring points of the air inlet and adjust the fan to make the wind pressure and air volume meter indicate 0.2 m³/s.
- Step 6: Make the experiment run continuously until the moisture sensor reading at each measurement point does not change and then end the experiment.
- Step 7: Repeat steps 1–5 and adjust the fan so that the airflow at the air inlet is 0.3 m³/s for data collection. Make the experiment run continuously until the moisture sensor reading at each measurement point does not change and then end the experiment.
- Step 8: Repeat steps 1–5 and adjust the fan so that the airflow at the air inlet is 0.4 m³/s for data collection. Make the experiment run continuously until the moisture sensor reading at each measurement point does not change and then end the experiment.

### 4. The Effect of Air Leakage on the Evaporation of Water from Goaf

In order to study the effect of air leakage on the evaporation of moisture in the goaf, this experiment was conducted by adjusting the stepless variable speed fan at the air outlet to achieve the variation in air leakage at the air inlet. The water evaporation was studied when the air leakage volume at the inlet was 0.02, 0.03 and 0.04 m³/s, respectively. The values of static pressure and wind speed at the air inlet are shown in Table 2 when the air leakage at the air inlet is 0.02, 0.03 and 0.04 m³/s, respectively. The data obtained from this

table were measured at measurement point 19, as shown in Figure 3b, using the Pitot tube as well as the wind speed and wind pressure and volume meter.

**Table 2.** Airflow parameters at the inlet.

| Serial Number | Air InletStatic Pressure Value/Pa | Wind Speed Valuem/s | Air Leakage Volumem$^3$/s |
|---|---|---|---|
| 1 | 6 | 2.1 | 0.02 |
| 2 | 10 | 3.2 | 0.03 |
| 3 | 15 | 4.2 | 0.04 |

### 4.1. Effect of Time and Air Leakage Variation on Coal Body Moisture

Figure 5a–c corresponds to the variation in moisture evaporation at air leakage volumes of 0.02, 0.03 and 0.04 m$^3$/s, respectively. At time 0 day, the moisture of the coal body at measurement points M1–M15 is around 30%. The main reason for the inconsistency in the coal moisture at measurement points M1~M15 is that the distribution and size of the coal particles around each measurement point are not exactly the same and there is a certain error. In addition, it can be seen from the figure that the coal moisture at all measurement points gradually decreases with the increase in evaporation time. The moisture of the coal body decreases from about 30% to about 7%. Under different air leakage volumes, the coal moisture of all three groups of experiments decreased rapidly in the initial 4 days' time. After 4 days, the rate of coal moisture reduction gradually decreased and then stabilized. The time from the beginning to the stabilization of the coal moisture was 15, 11 and 7 days for the air leakage volumes of 0.02, 0.03 and 0.04 m$^3$/s, respectively. Thus, it can be concluded that the larger the air leakage amount at the air inlet, the more moisture the air flow takes away from the model of the goaf, the faster the coal moisture evaporates and the shorter the time for the coal moisture to stabilize.

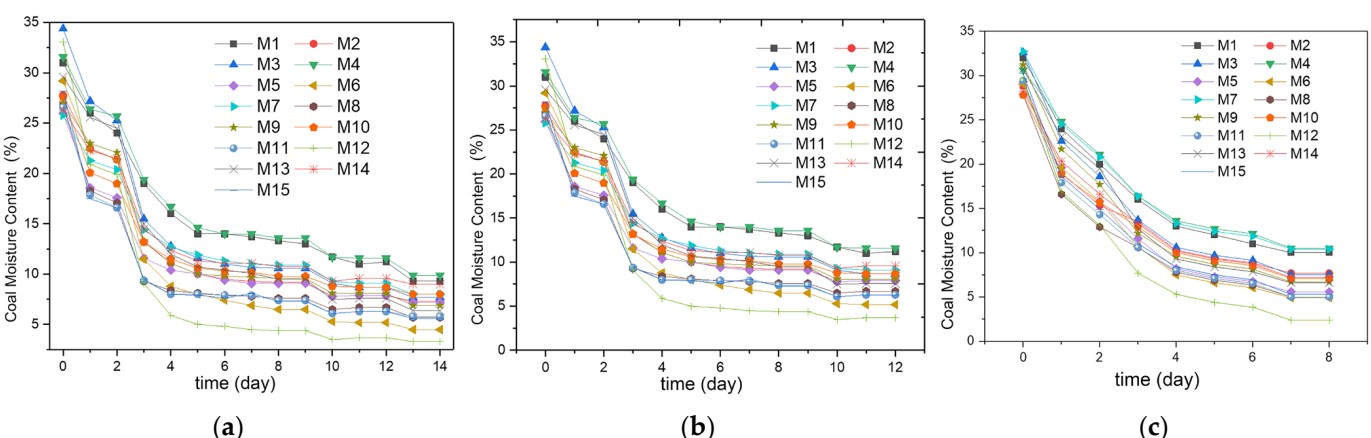

**Figure 5.** Variation in water evaporation with different air leakage volumes: (**a**) Change in water evaporation when the air leakage is 0.02 m$^3$/s; (**b**) Change in water evaporation when the air leakage is 0.03 m$^3$/s; (**c**) Change in water evaporate on when the air leakage is 0.04 m$^3$/s.

From the process of moisture reduction to stabilization, Figure 6a,b mainly depicts the moisture evaporation volume and moisture evaporation rate at different coal body moisture measurement points when the inlet air leakage volume is 0.02, 0.03 and 0.04 m$^3$/s, respectively. From the figure, it can be seen that the moisture evaporation amount and moisture evaporation rate at measurement point 12 are the greatest when the air inlet leakage is certain. Additionally, the water evaporation amount and water evaporation rate at measurement point 14 are the smallest. For the air inlet leakage of 0.02, 0.03 and 0.04 m$^3$/s, the moisture evaporation rates at measurement point 12 were 27.04, 29.3 and 29.41%, and the moisture evaporation rates at measurement point 14 were 18.29, 16.69 and 21.03%, respectively. Similarly, the moisture evaporation rates at measurement point 12 were 1.94,

2.67 and 3.90%/day, and those at measurement point 14 were 1.3, 1.52 and 3.00%/day. In addition, taking the air inlet leakage volume of 0.04 m³/s as an example, the moisture evaporation rates at different measurement points were ranked; then, the moisture evaporation rate at measurement point 12 was the largest, reaching 3.90%/day. The rate of water evaporation was 3.59%/d at measurement point 3, followed by measurement points 5, 6, 9, 11, 13 and 15, wherein the evaporation rate was around 3.5%/day. The rate of water evaporation was at the same level. Among these measurement points, the relatively large moisture evaporation rates were found at measurement points 6, 9, 11 and 15. The very small moisture evaporation rates were found at measurement points 4, 10 and 14, with 2.88, 2.95 and 2.99%/day, respectively.

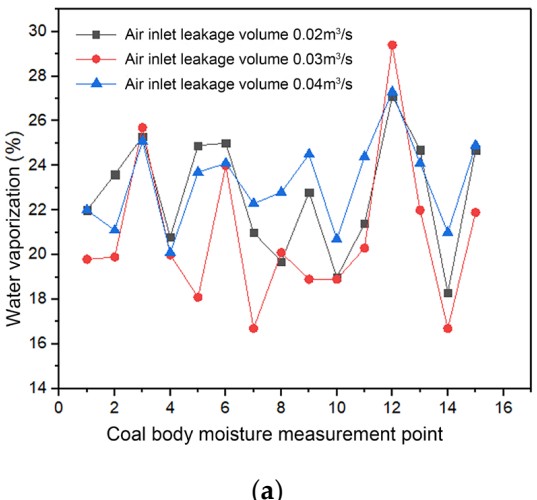

(**a**)

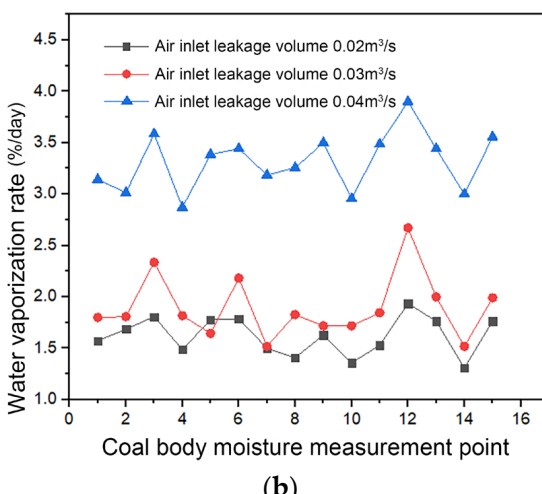

(**b**)

**Figure 6.** Amount of moisture change at different measurement points: (**a**) Water evaporation at different measuring points; (**b**) Water evaporation rate at different measuring points.

### 4.2. Influence of Measurement Point Location on the Amount of Water Evaporation

The coordinate axes xoy are defined for the bottom surfaces of the squares in Figure 6a,b, where the broad side axis of the three-dimensional model is the x-axis, and the long side axis of the three-dimensional model is the y-axis. Then, the coordinates of each measurement point are shown in Table 3.

**Table 3.** Plane coordinates of each measurement point.

| Measurement Points | x (cm) | y (cm) |
|---|---|---|
| 1 | 15 | 20 |
| 2 | 30 | 20 |
| 3 | 45 | 20 |
| 4 | 15 | 40 |
| 5 | 30 | 40 |
| 6 | 45 | 40 |
| 7 | 15 | 60 |
| 8 | 30 | 60 |
| 9 | 45 | 60 |
| 10 | 15 | 80 |
| 11 | 30 | 80 |
| 12 | 45 | 80 |
| 13 | 15 | 100 |
| 14 | 30 | 100 |
| 15 | 45 | 100 |

Based on the above rules and combined with Figure 3b, it can be concluded that both measurement points 12 and 15 are closest to the air inlet. However, the air flow through measurement point 12 is larger than that of measurement point 15. Different colors are used for plotting in Figure 7 to facilitate clear viewing. From Figure 7c, it can be seen that the relative humidity of the air at the air inlet is less than the relative humidity of the air at the air outlet. Therefore, the air flowing out of the goaf model will take away a large amount of moisture, so the moisture at measurement point 12 will be taken away first. Then, the evaporation rate of moisture at that place is the greatest. Additionally, most of the air flows away from the air outlet, so the moisture evaporation at measurement point 3 is only second to the moisture evaporation rate at measurement point 12. Then, the moisture evaporation rate is lower at measurement points 6, 9, 11 and 15. Combined with Figure 7a–c, the measurement points 3, 6, 9, 12 and 15 with higher moisture evaporation rates are all located in the first row, which is equivalent to the location near the working face in the goaf. Based on this rule, we also found that the water evaporation rate of the measurement points 2, 5, 8, 11 and 14 located in the second row is generally lower than that of the measurement points in the first row. The smallest evaporation rates correspond to the measurement points 4, 10 and 14, where the measurement points 4 and 14 are located in the third row. From this, we can derive the basic rule that, the closer the measuring point in the goaf is to the working face, the greater its moisture evaporation rate; the closer the measuring point in the goaf is to the location of the inlet and outlet, the greater its moisture evaporation rate.

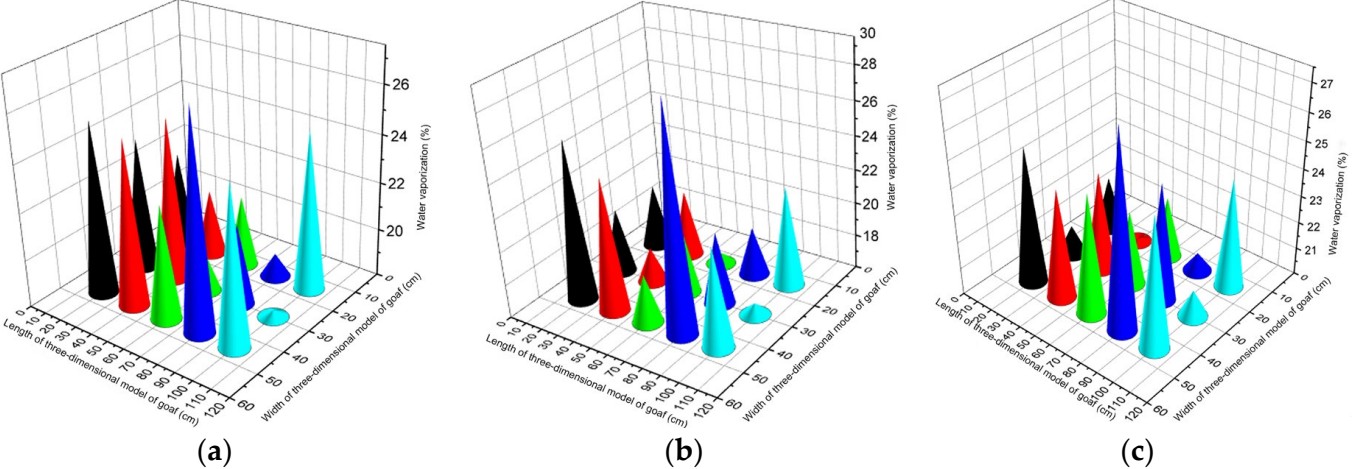

**Figure 7.** Evaporation of water without air leakage: (**a**) Diagram of water evaporation when the air leakage volume is 0.02 m³/s; (**b**) Diagram of water evaporation when the air leakage volume is 0.03 m³/s; (**c**) Diagram of water evaporation when the air leakage volume is 0.04 m³/s.

*4.3. The Effect of Time and Air Leakage Volume Changes on the Relative Humidity of the Air Inlet and Outlet*

The first two components are based on the amount of air leakage, the location of the measurement point and the time taken for the moisture at the measurement point to reach equilibrium as key factors. The first two sections summarize the patterns of water evaporation and water evaporation rates within the mining area model. The following is a further in-depth analysis of the reasons for the above law. In Figure 3b, the two measurement points 16 and 18 are used to determine the relative humidity of the air at the air inlet and air return. The changes in air relative humidity at the air inlet and outlet with different air leakage volumes are shown in Figure 8a–c.

From Figure 8a–c, the air relative humidity at the inlet and outlet shows irregular changes with time, and the change law of air relative humidity at the inlet is always consistent with the change law of air relative humidity at the outlet. The main reason for the change in the relative humidity trend with time and chaos is affected by the weather.

If the relative humidity of the air varies from day to day, the relative humidity of the air at the air inlet varies irregularly. However, there is an obvious law that the air relative humidity at the air inlet is always smaller than the air relative humidity at the air outlet. This means that the air flowing from the air inlet through the model of the goaf will take away part of the moisture, which leads to an increase in the relative humidity of the air at the air outlet. At the beginning of the experiment, the relative humidity difference between the air inlet and outlet was the largest. As time changes, the moisture evaporates and is taken away, and the relative humidity difference between the air at the inlet and outlet decreases. When the inlet air leakage volumes are 0.02, 0.03 and 0.04 $m^3/s$, the humidity differences at the inlet and outlet at the initial moment of the experiment are 16.49, 17.06 and 14.08%, respectively. When they reached equilibrium after 15, 11 and 7 days for the above three groups of coal body moisture evaporation experiments, the relative humidity differences of the air at the inlet and outlet were 4.05, 5.03 and 1.99%, respectively. Finally, the fitted curves of the air relative humidity difference at the inlet and outlet with time at the inlet leakage volumes of 0.02, 0.03 and 0.04 $m^3/s$ are y = $-0.89 \times$ x + 15.12, $R^2$ = 0.89; y = $-1.05 \times$ x + 16.21, $R^2$ = 0.83; y = $-1.58 \times$ x + 11.45, $R^2$ = 0.78, respectively. It can be seen that, as the air inlet leakage increases, the greater the rate of change of the air relative humidity difference becomes at the air inlet and outlet. Additionally, the faster the water evaporates from the coal body in the mining area, the shorter the time it takes to reach equilibrium. The corresponding air relative humidity difference change rates are 0.89, 1.05 and 1.58%/d.

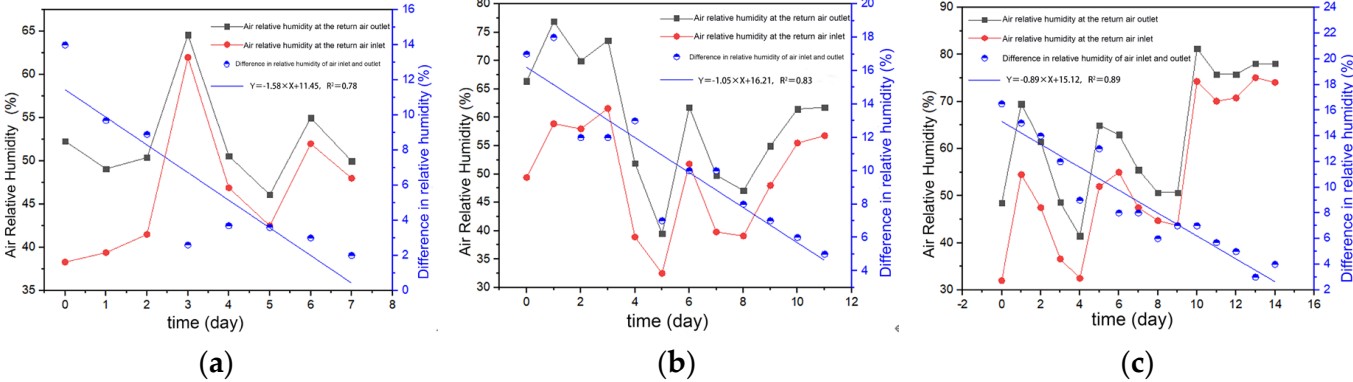

**Figure 8.** Variation in air relative humidity at air inlet and outlet with different air leakage volumes: (**a**) Relative humidity of the air inlet when the air leakage volume is 0.02 $m^3/s$; (**b**) Relative humidity of the air inlet when the air leakage volume is 0.03 $m^3/s$; (**c**) Relative humidity of the air inlet when the air leakage volume is 0.04 $m^3/s$.

## 5. Conclusions

Through the observation and analysis of the three-dimensional model data, we know that, as the air leakage volume increases, the rate of change of the relative humidity difference between the air inlet and outlet increases, the evaporation of coal body moisture in the goaf is accelerated and the time to reach coal body moisture equilibrium becomes shorter. Therefore, the phenomenon presented is that, when the air inlet leakage increases, the wind flow takes away more moisture from the model of the goaf, the coal body moisture evaporation accelerates and the coal body moisture stabilization time becomes shorter; the closer the measuring point in the goaf is to the working face, the greater the moisture evaporation rate; the closer the measuring point in the goaf is to the inlet and outlet, the greater the moisture evaporation rate. Therefore, we can conclude that, the closer the area to the working face, the more obvious the failure of the inhibitor at the same time; and in these areas, the closer the area to the inlet and outlet, the more obvious the failure of the inhibitor in the same time. The model is used to study the change of coal moisture inside the goaf under different air leakage volumes and to avoid personnel entering the complex goaf to collect data such as coal moisture.

**Author Contributions:** Conceptualization, C.C.; methodology, C.C.; formal analysis, C.C.; investigation, C.C.; resources, S.J.; data curation, Z.J.; writing—original draft preparation, Z.J.; writing—review and editing, Y.Z., Y.Y., J.L. and Z.S.; supervision, S.J. All authors have read and agreed to the published version of the manuscript.

**Funding:** This research was funded by the Natural Science Foundation of Shanxi Province (No. 20210302124134), the National Natural Science Foundation of China (Nos. 52204228, 52074285) and the China Postdoctoral Science Foundation (No. 2021M690530).

**Institutional Review Board Statement:** Not applicable.

**Informed Consent Statement:** Not applicable.

**Data Availability Statement:** Not applicable.

**Conflicts of Interest:** The authors declare no conflict of interest.

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
