# Peer review of "Experimental Study of Three-Dimensional Time Domain of Water Evaporation from Goaf"

_applsci, doi:10.3390/app13137932_

Round 1
Reviewer 1 Report
This paper investigated a three-dimensional time domain of water evaporation from goaf. While reviewing this manuscript, I focused on the structure, the content, coherence, grammar, typos, etc. I was impressed with the structure and the analysis of the report. As a reviewer, some general comments and suggestions pertaining to this manuscript are as follows.
Recent studies related to coal oxidation, coal moisture content, factors affecting coal spontaneous combustion and effects of chemical inhibitors on coal spontaneous combustion have been carried out by many scholars. These scholars have reported their findings on effects of moisture on spontaneous combustion of coal. However, the authors of this paper only limited this study to scientists in a particular region. I suggest the authors should introduce recent articles related to the subject and incorporate them in the paper.
The abstract should be written to reflect the scientific background and findings of the study.
The novelty of this study is not well captured in the report.
Some of the sentences used in the whole report are too long, please make them concise and check punctuation marks accordingly.
What test standards (procedures) were used for the proximate and ultimate analysis and why were they not cited accordingly?
The results and discussion are meant to provide a clear interpretation of findings and should be based on observation as well as scientific facts “literature” and “reasons”.
Authors should check the caption of Table 2.
Figure 6a-c is blurred.
The conclusion should be written in a concise model.
With regards to the raised comments, the manuscript should be accepted with minor corrections.
Author Response
请参阅附件。

Reviewer 2 Report
The manuscript presents the results of experimental measurements of water evaporation from goaf. The study is novel and presents interesting results; however, I have several concerns with the paper.
1. The English grammar is not terrible, but it needs to be corrected and improved.
2. The rationale behind the experimental layout (location of air inlet and outlet and dimensions of bed) needs to be explained, including how this layout connects to real coal mine geometries.
3. Your first conclusion states that higher air flow rates removes moisture more quickly. This is a simplistic and obvious conclusion. We probably didn't need experiments to show this. This needs a deeper analysis.
4. Similarly, your second conclusion states that evaporation is higher closer to the working face. This is also rather obvious, since this is where air flow rates will be highest. Is there deeper meaning?
5. The Introduction suggests these studies are needed to improve mine safety. But the paper does not connect the results of the study to any conclusions or insights that could be used to guide or improve mine practices to enhance safety. Therefore, I am left with no answer to the key question, "What is the value of this study?"
The English in the manuscript is better than in other manuscripts I have reviewed; however, there are still grammatical errors that need to be addressed.
